# 3D Model of the Early Melanoma Microenvironment Captures Macrophage Transition into a Tumor-Promoting Phenotype

**DOI:** 10.3390/cancers13184579

**Published:** 2021-09-12

**Authors:** Gabriela A. Pizzurro, Chang Liu, Kathryn Bridges, Amanda F. Alexander, Alice Huang, Janani P. Baskaran, Julie Ramseier, Marcus W. Bosenberg, Michael Mak, Kathryn Miller-Jensen

**Affiliations:** 1Department of Biomedical Engineering, Yale University, New Haven, CT 06511, USA; gabriela.pizzurro@yale.edu (G.A.P.); chang.liu.cl887@yale.edu (C.L.); kate.bridges@yale.edu (K.B.); amanda.alexander@yale.edu (A.F.A.); alice.huang@yale.edu (A.H.); janani.baskaran@yale.edu (J.P.B.); 2Department of Dermatology, Yale University, New Haven, CT 06511, USA; JRamseier@mednet.ucla.edu (J.R.); marcus.bosenberg@yale.edu (M.W.B.); 3Department of Pathology, Yale University, New Haven, CT 06511, USA; 4Department of Immunobiology, Yale University, New Haven, CT 06511, USA; 5Department of Molecular, Cellular and Developmental Biology, Yale University, New Haven, CT 06511, USA

**Keywords:** melanoma, tumor microenvironment, tumor-associated macrophages, fibroblasts, type I collagen, 3D culture, immunosuppression

## Abstract

**Simple Summary:**

We developed a “tumor-in-a-dish” experimental system to study the early events favoring tumor growth and suppression of the immune response in metastatic melanoma. We combined murine melanoma tumor cells with fibroblasts and macrophages in a 3D collagen matrix and characterized how interactions between these three cell types, which are present in the early stages of tumorigenesis, drive immune suppression and the tumor-promoting transition in macrophages that is observed in vivo. Over the course of 7 days in the co-cultures, we quantified the dynamics of cues transmitted by direct cell–cell interactions, through the extracellular matrix and through secretion of immune mediators. We found that macrophages acquired features and a functional profile consistent with those present in in vivo murine melanoma tumors. This system will enable future studies of macrophage–stromal cross-talk in the melanoma microenvironment and provide a platform to test potential therapeutic approaches aimed at stimulating immune activity in macrophages.

**Abstract:**

Tumor immune response is shaped by the tumor microenvironment (TME), which often evolves to be immunosuppressive, promoting disease progression and metastasis. An important example is melanoma tumors, which display high numbers of tumor-associated macrophages (TAMs) that are immunosuppressive but also have the potential to restore anti-tumor activity. However, to therapeutically target TAMs, there is a need to understand the early events that shape their tumor-promoting profile. To address this, we built and optimized 3D in vitro co-culture systems, composed of a collagen-I matrix scaffolding murine bone-marrow-derived macrophages (BMDMs), YUMM1.7 melanoma cells, and fibroblasts to recreate the early melanoma TME and study how interactions with fibroblasts and tumor cells modulate macrophage immune activity. We monitored BMDM behavior and interactions through time-lapse imaging and characterized their activation and secretion. We found that stromal cells induced a rapid functional activation, with increased motility and response from BMDMs. Over the course of seven days, BMDMs acquired a phenotype and secretion profile that resembled melanoma TAMs in established tumors. Overall, the direct cell–cell interactions with the stromal components in a 3D environment shape BMDM transition to a TAM-like immunosuppressive state. Our systems will enable future studies of changes in macrophage–stromal cross-talk in the melanoma TME.

## 1. Introduction

The immune response against tumors is conditioned by the tumor microenvironment (TME), which is often highly immunosuppressive and depleted of effector cells [1]. In the TME, cells communicate via direct contact and a network of secreted signals established early during tumor development. Non-tumor cell types within the milieu co-evolve with tumor cells, disrupting tissue homeostasis while favoring disease progression and metastasis [2]. Thus, cell–cell interactions between immune cells, tumor cells, and other support cells play an active role in shaping the immune environment in tumor progression and responses to immunotherapy [3].

Melanomas are very aggressive tumors and frequently lethal, but they have been particularly responsive to immune checkpoint inhibitors aimed at boosting T cell function, specifically through PD-1 and CTLA-4 blockade [4,5]. On the other hand, advanced melanomas usually display low infiltration of T cells and high numbers of tumor- associated myeloid cells (TAMs), and the latter correlates with a worse prognosis [6]. TAMs and cancer-associated fibroblasts (CAFs) are key players in the TME, providing support for both tumor and immune cells depending on the context [3,7]. TAMs reprogram their functional state as the tumor grows, although the specific mechanisms leading to the accumulation of immunosuppressive TAMs remain unclear. This phenotypic and functional plasticity make TAM-targeted therapies a promising treatment option [8,9,10]. The currently available data are not sufficient to explain or predict treatment outcomes, since the immune activity in the TME is an emergent property of many interacting cell populations [11].

In recent years, the biomechanical interactions within the TME have been of interest since this interplay can induce and regulate specific functional phenotypes of its components. Cells in the TME are embedded in an extracellular matrix (ECM), which has a complex and evolving composition and structure and mediates a series of dynamic physical interactions between tumor and stromal cells [12]. Collagen structure can modulate TAM polarization and changes in macrophage morphology have suggested different functional phenotypes, such as higher expression of arginase (Arg)-1, CD206 and Chi3l3 [13]. Higher collagen density usually correlates with higher numbers of immunosuppressive macrophages and the reinforced confinement can down-regulate pro-inflammatory cytokine secretion [14,15]. Increased collagen stiffness can regulate immune cell spreading, motility and response to treatment [16,17]. Fibroblasts are key remodelers of the ECM, through secretion of ECM proteins, proteolysis via metalloproteinases, and force-mediated ECM reorganization [18,19]. Dynamic forces in the ECM have a role in initiating and guiding macrophage migration. As the major source of contractile forces, fibroblasts cue and recruit macrophages from hundreds of micrometers away [20].

Intercellular signaling between TAMs and other cell types is central to establishing and maintaining an immunosuppressive TME and is a critical target to reestablish an effective immune response [21,22]. Although identifying and targeting immuno-suppressive TAM subpopulations based on single markers, such as CD163 or Arg-1, has shown positive results [23,24,25], developing these approaches remains challenging due to their variation between species and tumor types [26,27,28]. TAMs have also been associated with resistance to checkpoint inhibitors treatment [29]. However, targeting TAMs through the combination of CD40 agonist and CSF1R inhibitor induces a new pro-inflammatory TAM subpopulation that leads to a change in the TME and T-cell dependent tumor control [21]. Determining how TAMs communicate with other cell types in the melanoma TME is needed to develop more effective methods to target the network of extracellular signals [30,31].

Studying all these variables in dynamic in vivo environments can be challenging when trying to identify individual contributions to TME evolution. Recent studies have demonstrated that organotypic melanoma cultures outperform 2D assays when studying TME-imprinting mechanisms and closely resemble tumor growth as observed in human lesions while supporting cell survival and function [32,33]. The use of 3D cultures may accelerate the identification of predictive and/or prognostic markers and the development of effective combination therapies [34]. Ex vivo systems that incorporate key features of the native TME and model the dynamic response to checkpoint inhibitors can facilitate efforts in precision immuno-oncology.

In the present study, we developed a simple system of early tumor formation to recreate the environment observed in vivo, and describe the interactions between macrophages, fibroblasts and tumor cells that occur during this process. In this setting, we addressed questions on the specific roles of their components and mediators driving the generation of the early immunosuppressive melanoma TME. Our work recapitulated the evolution of the dynamic interactions in an in vitro 3D co-culture model. Specifically, we demonstrated how profiles of cytokines evolve over time and also how phenotype, morphology, and migration of macrophages change longitudinally. This work provides insights on early cell–cell interactions in the melanoma microenvironment.

## 2. Materials and Methods

### 2.1. Animals

Female C57BL/6J and B6.129S4-Arg1tm1.1Lky/J Arg-1 reporter mice of 6–8 weeks of age were purchased from Jackson Laboratories. Mice were housed according to the standard housing conditions of the Yale Animal Resources Center in specific pathogen-free conditions. Mice were left to acclimate for one week before use. All animal experiments were performed according to the approved protocols of the Yale University Institutional Animal Care and Use Committee.

### 2.2. Cell Lines

Yale University Mouse Melanoma (YUMM) 1.7 and YUMM1.7 exposed to radiation (YUMMER) cell lines, both WT and GFP^+^, were kindly provided by Marcus Bosenberg, Yale University [35,36]. 3T3MEFs WT (CRL-2742) was acquired from ATCC. 3T3MEF-tdTomato cells were previously generated in the lab using a tdTomato vector transfection (0036VCT, Vectalys, Takara, France) and selected for stable expression of the fluorescent protein, according to the manufacturer’s instructions. Cells were grown in DMEM/F12 media supplemented with 10% FBS, 1% NEAA, 2 mM L-glutamine and 1% sodium pyruvate. 3T3MEFs were treated with 10 ng/mL TGF-β1.

### 2.3. BMDM Culture

Bone marrow-derived macrophages (BMDMs) were generated as previously described [37]. Briefly, bone marrow was extracted from the hind-leg bones of the mouse by the flushing method. After red blood cell lysis with ammonium-chloride-potassium lysis buffer (ACK Lonza, Walkersville, MD, USA) cells were incubated for 4 h at 37 °C with 5% CO_2_ in a non-tissue culture (TC)-treated plastic petri dish with BMDM media (RPMI supplemented with 10% FBS, 100 U/mL penicillin, 100 g/mL streptomycin, 1% sodium pyruvate, 25 mM HEPES buffer, and 50 M 2-mercaptoethanol). After 4 h, the non-adherent cells were transferred to a new petri dish and incubated with BMDM media + 20 ng/mL macrophage-colony stimulating factor (M-CSF; Peprotech, Rocky Hill, NJ, USA). After 3 days, an additional 10 mL of BMDM media + 20 ng/mL M-CSF was added to the plate. After a total of 6 days, BMDMs were harvested in phosphate-buffered saline (PBS) + 5 mM ethylene-diaminetetraacetic acid (EDTA) with gentle pipetting, and the cells were ready for further use.

### 2.4. BMDM Polarization

BMDMs were plated non-TC treated multi-well plates (Falcon, Agawam, MA, USA) at a density of 100,000 cells/cm^2^ and cultured in BMDM media + 20 ng/mL M-CSF. For polarization, cells were stimulated for 24 h with 100 ng/mL lipopolysaccharide (LPS) (LPS-EK Ultrapure, InvivoGen, San Diego, CA, USA) and 10 ng/mL IFN-γ (Peprotech, Rocky Hill, NJ, USA) for M1 profile, or 20 ng/mL IL-4 (Peprotech, Rocky Hill, NJ, USA) for M2 profile.

### 2.5. Tumor Studies and Sample Processing

Female C57BL/6J mice were anesthetized with isoflurane and intradermally injected with 3.5 × 10^5^ tumor cells in both flanks. Mice were monitored every other day for tumor growth. Tumor volume was assessed by measuring caliper tumor length (L) and width (W) and calculated as 0.52 × L × W^2^. At the designated timepoints, mice were euthanized in a CO_2_ chamber and tumors were resected and weighted for processing. Briefly, tumors were first cut with scissors and then chopped with razor into 1 mm^3^ pieces. They were transferred to a tube with 10 mL of digestion buffer (1X PBS Ca^+^Mg^+^ containing 0.1 mg/mL DNase I, Roche 05401127001, and 0.82 mg/mL Collagenase IV, from C. histolyticum, Sigma-Aldrich C1889, St. Louis, MO, USA) and incubated at 37 °C in shaker for 30 min. Samples were vortexed, filter in 40 µm cell strainer and placed on ice. Cells were washed and resuspended in ACK lysis buffer at RT for 5 min. Samples were filtered and washed again, resuspended and counted.

### 2.6. Immunostaining, Flow Cytometry and Cell Sorting

For flow cytometry, we used the following antibodies and dyes (clone, cat): CD45 AF700 (30-F11, 103128), CD45 PerCP (30-F11, 103132), CD11b BV421 (M1/70, 101236), F4/80 PerCP (BM8, 123126), CSF1R BV605 (AFS98, 135517), Ly6C BV711 (HK1.4, 128037), Ly6C AF488 (HK1.4, 128022), Ly6G AF647 (1A8, 127610), iNOS AF488 (CXNFT, 53-5920-82) and APC (CXNFT, 17-5920-82), CD40 PE (3/23, 124609), CD86 PE-Dazzle594 (GL1, 105042), MHCII APC-Cy7 (M5/114.15.2, 107628), CD206 PE-Cy7 (CO68C2, 141720), Arg1 APC (A1exF5, 12-3697-82), and Live/Dead eFluor506 (423101) BioLegend. For some of these experiments, we used BMDMs from Arg1-YFP reporter mice. For phenotype analysis, at the day for processing, cells were obtained from the tumors, as previously described, or retrieved from the collagen gel by digestion with Collagenase (Advanced BioMatrix 5030, San Diego, CA, USA) following the manufacturer’s manual. Single-cell suspensions were stained in FACS Buffer (PBS 2% FBS). Briefly, cells were incubated with FcBlock 1/200 (anti-CD16/CD32, eBiosciences, San Diego, CA, USA) for 20 min at 4 °C. Cells were washed and then incubated with the antibody mix for extracellular markers. For intracellular staining, we used the CytoFix/CytoPerm and Perm/Wash Buffer kit (BD, 554714), according to the manufacturer’s instructions. Cells were then incubated with the antibody mix for intracellular markers. Samples were resuspended in 500 L PBS and were analyzed on a LSRFortessa (BD Biosciences, San Jose, CA, USA). For analysis, samples were gated for singlets and live cells. Macrophages were gated on CD11b^+^F4/80^+^ (low and high) population. For TAM sorting, single-cell suspensions from tumors were prepared as described above, and stained without fixing. TAMs were processed in a FACSAria instrument (BD Biosciences) and sorted as singlets, Live^+^, CD45^+^CD11b^+^Ly6G^−^. For sorting cells from the 3D co-culture, single cells suspensions were sorted from Live^+^ cells: GFP^+^YUMM cells, CD45^+^CD11b^+^ BMDMs and negative selection for 3T3MEFs. Flow cytometry data were analyzed using FlowJo software (TreeStar Inc, Ashland, OR, USA).

### 2.7. Multiplex Protein Secretion

For monitoring the evolution of the microenvironment in the 3D cultures, 100 µL were collected from the media in each well at indicated timepoints. For all the rest of the different sample types, cells were plated at a density of 1 × 10^6^/mL in cell culture plates and incubated supernatants from in vitro and ex vivo cultures were collected, centrifuged for removing cell debris and kept at −80 °C until further processing. They were then submitted to Eve Technologies Corp (Calgary, AB, Canada) to perform a Mouse Cytokine/Chemokine Array 44-Plex (MD44) assay, which is based on color-coded polystyrene beads combined with a dual-laser and a flow-cytometry system for sample acquisition and analysis. All detected analytes were within the dynamic range of the standard curves of each analyte, observing no saturation in the samples analyzed. For data analysis purposes, those presenting an out of range (OOR) measurement below the parameter logistic standard curve were systematically replaced with the lowest value obtained for a particular analyte, as per the suggestion of the company. The data were then either natural log-transformed or converted to z-scores to aid in visualization. Samples were visualized and hierarchically clustered using the clustermap function from the Seaborn module in Python.

### 2.8. Single-Cell Secretion Assay and Analysis

The single-cell secretion profiling experiments were performed as described [38]. TAMs were loaded into the Polydimethylsiloxane (PDMS) microwells in BMDM media, and exposed to the flow-patterned antibody array slide for Chi3l3, MMP9, IGF-I, IL-10, CCL17, CCL22, CCL2, CCL3, CCL5, IL-27, IFN-β1, CXCL1, TNF-α, IL-6, IL-12p40, and analyzed as previously described. For visualization, single-cell secretion data were embedded in two-dimensional space with Potential of Heat diffusion for Affinity-based Transition Embedding (PHATE) [39]. This dimensionality reduction algorithm better preserves the nonlinear progressions and branching that describe the spectrum of continuous phenotypes expected in single-cell data from a single cell type. To investigate functional heterogeneity, cluster analysis of the single-cell secretion data was accomplished with PhenoGraph [40], which is an unsupervised, graph-based clustering method developed to identify subpopulations in high-dimensional single-cell data. Extracted clusters were analyzed using custom scripts written in Python. PHATE and PhenoGraph are both available as publicly available software packages in Python.

### 2.9. Histology and Immunofluorescence

Tumor samples were fixed in formaldehyde and paraffin-embedded. Immunohistochemistry for EGFR was performed on tumors slices and counterstained. Hematoxylin/Eosin (H&E) staining was performed on tumor slices. Fibroblast density was calculated as area per High-Power Field (HPF). For immunofluorescence on 3T3MEFs, cells were plated onto round, poly-L-lysine-coated glass coverslip in 24-well plates. YUMM conditioned media (CM) was generated by culturing cells to 100% confluency and then replacing media for FBS-free RPMI. CM was collected after 48 h, fractioned and at −80 °C. Cells were left unstimulated, or treated with 5 ng/mL TGF-β1 (Peprotech, Rocky Hill, CY, USA) or YUMM conditioned media, for 24 h. Then cells were fixed in methanol, stained with anti-SMA AF488 (ab184675, Abcam, Cambridge, UK) and anti-FAP (ab28244, Abcam, Cambridge, UK) overnight at 4 °C. Then they were incubated with Goat anti-Rabbit PE and Hoechst. Coverslips were mounted and analyzed in a fluorescent microscope.

### 2.10. Stained Collagen Pulling Assay

Rat tail type I collagen (Corning, New York, NY, USA) was labeled with Alexa Fluor 647 NHS Ester (Succinimidyl Ester) and dialyzed as before [41]. The CM was then made by diluting thawed supernatants 1:2 with fresh culture media. In the assay, 3T3MEFs (tdTomato) were embedded in 1.5 mg/mL stained collagen at a density of 600 K/mL. Cells were treated with fresh media or YUMM CM, added after 1 h gelation at 37 °C and replaced daily. Confocal tilescan imaging was taken on day 5.

### 2.11. Proliferation Assay

Once BMDMs were differentiated and stimulated, they were stained with 5 uM CFSE (Molecular probes, Eugene, OR, USA, Life Technologies, Carlsbad, CA, USA) in PBS. In brief, cells were washed, resuspended with the CFSE solution to 2 × 10^6^/mL and stained for 10 min at 37 °C, plus 5 min on ice. Then, an equal volume of complete media was added, and left for 5 min at room temperature. Cells were then centrifuged at 1500 rpm for 5 min, resuspended in fresh media and counted for subsequent use. Analysis of CFSE assay was performed using FlowJo software (TreeStar, Woodburn, OR, USA).

### 2.12. Spheroid Assay

BMDMs, 3T3MEFs and YUMM cells were first stained with CellTracker™Deep Red Dye(C34565), CellTracker™ Orange CMTMR Dye(C2927), and CellTracker™ Green CMFDA Dye(C7025), respectively, at 37 °C incubator for 45 min, then washed in fresh medium and resuspended in 2% Matrigel (Corning 354230). Resuspended cell solutions were aliquot in agarose gel-coated 96 well plates with a fixed density of 12 K BMDMs, 1 KYUMM cells, and 3 K 3T3MEFs per well. The cell-loaded plates were centrifuged with 1000 *g* for 10 min and then kept in 37 °C, 5% CO_2_ incubator. Spheroids were collected on day 4 and embedded in 1.5 mg/mL rat tail type I collagen (Corning CB354249) for further study.

### 2.13. 3D Cell Culture

BMDMs, 3T3MEFs, and cancer cells were harvested and seeded in neutralized 1.5 mg/mL rat tail type I collagen (Corning CB354249) as previously described [42]. Complete growth medium was added after 1 h gelation at 37 °C incubation. The 3D co-culture was maintained in 37 °C, 5% CO_2_ with medium changed every 1–2 days. Unless other noted, BMDMs, 3T3MEFs and cancer cells were seeded at a density of 2400 K/mL, 200 K/mL and 200 K/mL, respectively. On the day for processing, cells were retrieved by digestion with collagenase (Advanced Matrix 5030), following the manufacturer’s instructions.

### 2.14. Collagen Quantification

In the collagen quantification assay, BMDMs, 3T3MEFs and cancer cells were seeded at a density of 2400 K/mL, 600 K/mL and 200 K/mL, respectively, in a 96 well plate. Only half of the medium was changed daily and was gently added in a dropwise manner to minimize the pipetting disturbance to the collagen. On day 3 and day 7, the 96 well plates were frozen and kept at −80 °C immediately after medium removal. Fresh tumor samples were collected on day 7 and day 14 and kept frozen at −80 °C after mass and volume measurement. Collagen was measured using the SirCol Collagen Assay Kit (Accurate CLRS1111). Both in vitro culture and tumor samples were thawed right before the assay and processed according to the manufacturer’s instructions. To note, after acid extraction, tumor samples were first filtered by a 100 µm cell filter and then centrifuged at 5000 *g* for 10 min to remove residual undigested debris. The total collagen in the supernatants was measured and used to represent the total soluble collagen in the tumor sample.

### 2.15. Confocal Imaging

Confocal imaging was performed in a Leica TCS SP8 microscope (Leica Microsystems Inc., Buffalo Grove, IL, USA, 20X0.75NA objective) to track 3D culture along time. For these experiments, we used GFP^+^ melanoma cells. BMDMs were stained with CellTracker Deep Red Dye (C34565) and 3T3MEFs with CellTracker Orange CMTMR Dye (C2927) at 37 °C incubation for 45 min before co-culture. Alternatively, 3T3MEFs tdTomato were used for imaging. Then, 12 h time-lapse or tile scan images were collected on day 0, day 1, day 3, day 5, and day 7 for macrophage migration and morphology analysis.

### 2.16. Confocal Image Analysis

Macrophage migration was tracked with TrackMate on Fiji. Cell center was defined by averaged coordinates of traced cell boundary on a projected image, and average speed was calculated as the mean of the absolute value of the cell net displacement every 1 hr time intervals. Mean squared displacements were computed with the following equation:(1)$$\mathrm{MSD}\left(n\right)=\frac{1}{N-n+1}\overset{N-n}{\underset{i=0}{\sum }}\left[{\left(x_{i+n}-x_{i}\right)}^{2}+{\left(y_{i+n}-y_{i}\right)}^{2}\right]$$
where *x* and *y* indicate *x* and *y* coordinates, *n* indicates the *n*-th step and *N* total step number. For morphology analysis, high-resolution XYZ tile scan imaging was collected longitudinally. Imaging tiles were merged and projected before segmentation. Macrophage compactness, circularity and elongation index was calculated by following equations in MATLAB, respectively:(2)$$\mathrm{Compactness}=\frac{4\times Pi\times Area}{Perimeter^{2}}$$
(3)$$\mathrm{Circularity}=\frac{4\times Pi\times Area}{ConvexPerimeter^{2}}$$
(4)$$\mathrm{Elongation}=\frac{Width_{boundingbox}}{Length_{boundingbox}}$$

In the tumor–macrophage distance measurement, the xy coordinates of cell center were tracked by TrackMate while z coordinate was decided manually by identifying the slice with the highest intensity pixels and clearest cell boundaries. To calculate 3D distance at each time point is cumbersome, hereby we use the cell position at the beginning of imaging (immediately after 1 h gelation) to represent the paired 3D distance.

The average speed of macrophages was calculated in the same way as the above-mentioned by deriving the planar net displacement per hour. To properly define the close and far macrophages, each paired macrophage–tumor cell distance was ranked from smallest (close to nearest tumor cells) to largest (far from nearest tumor cells). The average speed of the fifteen closest macrophages (i.e., close macrophages) and the fifteen farthest macrophages (i.e., far macrophages) was calculated by taking the mean of the macrophage migration speed in each group respectively.

### 2.17. Statistical Analysis

Unless noted otherwise, one-way ANOVA or two-way ANOVA was used with post-hoc comparisons in Prism9. Principal component analysis was performed using SIMCA 16 (Sartorius, Göttingen, Germany). Pearson correlation coefficients were calculated for each pairwise combination of samples using the Pearson function from the scipy.stats module in Python. These results were visualized with Matplotlib’s matshow function. For far-close tumor–macrophage distance comparison, one-tailed t-test was performed to compare the average speed of close and far macrophages.

## 3. Results

### 3.1. In Vitro-Polarized BMDM Phenotypic and Functional Profiles Are Distinct from TAM Profiles

To study the early events in the melanoma TME that shape the TAM tumor-promoting profile, we proposed to build a 3D in vitro culture system using the Yale University Melanoma Model (YUMM) 1.7, which has genetic changes common to human melanomas, including BRAFV600E, Pten^−/−^, and Cdkn2^−/−^ [35,36]. In order to have a benchmark for our model of melanoma TAMs in vitro, we first isolated TAMs from YUMM1.7 murine tumors on days 7, 10 and 16 and characterized their phenotypic and functional profiles. The melanoma TAMs from YUMM1.7 (Y) and YUMMER1.7 (YR) tumors showed an overall similar phenotypic profile when contrasted against peritoneal macrophages isolated from the same mice, with slight variations across timepoints (Appendix A). However, these melanoma TAMs displayed a heterogeneous phenotype, clustering into multiple subsets expressing a combination of markers canonically associated with opposed polarization states. When we isolated TAMs from the tumor and cultured them in vitro, they quickly lost this complexity and became a more homogeneous population (Appendix A).

We next generated BMDMs with different polarization states, which we proposed to use to model TAMs in our in vitro co-cultures: unstimulated BMDMs (M0) and stimulated for 24 h with LPS+IFN-γ (M1) or IL-4 (M2). We compared their characteristics to the melanoma TAMs, evaluating the expression of CD11b, F4/80, Ly6C, CSF1R, iNOS, CD40, CD86, MHCII, Arg1, CD163 and CD206 expression by flow cytometry. We used principal component analysis (PCA) to visualize the expression of M0, M1, and M2 BMDMs versus TAMs isolated from melanomas, and we found that TAMs showed an intermediate state (Figure 1a). Interestingly, we also found that TAMs grouped by the stage of the tumor, early (day 7) or late (day 14), rather than by the tumor from which they were isolated. When the single-cell FACS data were visualized using UMAP, we found that in vitro-polarized BMDMs had a more consistent expression of canonical markers and were more tightly clustered than the TAMs, which appeared to be composed of subpopulations with a mixed expression of M1- and M2-associated markers (Figure 1b,c).

We also defined functional aspects of the melanoma TAMs in the TME through their secretion profiles. Again using PCA to visualize the population-level secretion program for each group, we observed that TAMs occupied a distinct functional state from M0, M1, and M2 BMDMs, and these differences increased over time (Figure 1d). Analysis of the cytokine and chemokine (C/C) secretion levels showed that TAMs secrete a broader number of C/Cs than BMDMs with well-defined polarization states (Figure 1e), and displayed more variance between samples (Appendix A). TAMs isolated from day 14 tumors were more active secretors than those from day 7 tumors.

To further dissect the functional subsets of melanoma TAMs, we studied their single-cell secretion using a multiplex panel for M1/M2-like profiling in an in-house microwell device, developed in our lab (Appendix A), Ref. [43] and previously used to characterize BMDMs and TAMs [21,38]. We found that approximately 60% of TAMs isolated from these melanoma tumors were not actively secreting any of the measured proteins (Appendix A), similar to previous observations [21]. When we compared single-cell secretion from BMDMs polarized in vitro [38] to melanoma TAMs, we found that TAMs displayed a secretion profile that placed them between the M1 and M2 in vitro-defined profiles (Figure 1f,g), similar to our observations of their phenotypic profiles (Figure 1b,c). By performing clustering analysis on the single-cell secretion and visualizing them in PHATE, we identified functional clusters within these melanoma TAMs, which aligned into two activation axes, aside from the low-secreting subset. One cluster of TAMs defined a more immunosuppressive axis, with a more committed M2-like profile, characterized primarily by the secretion of Chi3l3, with a fraction of them co-secreting TNF-α. The other axis exhibited a mixed pro-inflammatory profile, more clearly defined by the secretion of TNF-α. One main cluster predominantly secreted this cytokine with a fraction co-expressing IL-10, and a second cluster, with more complex polyfunctional features (Figure 1h and Appendix A). TAMs from both melanoma tumors exhibited similar profiles of secretion, with single-cell functional sub-clusters heterogeneity (Appendix A). Overall, we conclude that melanoma TAMs from growing tumors exhibit mixed M1/M2 profiles, both at the population and single-cell level, that is distinct from M1 and M2 polarization states generated from simple in vitro cultures.

### 3.2. 3D Collagen Cultures with BMDMs Generate Dynamic Systems Appropriate to Model the Early Melanoma TME

When the melanoma cells are inoculated in the mouse, they come in contact with the dermal components, such as fibroblasts, immune and endothelial cells, and ECM, and infiltrate through the collagen structure (Figure 2a). Between day 5 and day 7, fibroblast infiltration changes from being evenly distributed across different regions of the tumor to being isolated in the periphery (Figure 2b and Appendix A). Between days 7 and 14, YUMM melanoma tumors exhibited consistent growth, leading to a significant increase in tumor volume (Figure 2c left and Appendix A). To determine if this early change in fibroblast infiltration could have a sustained impact on the composition of the ECM and its interaction with the other TME components as tumor continued to grow, we quantified the soluble collagen present within the tumor structure of this melanoma model between days 7 and 14. In YUMM tumors, the collagen concentration significantly decreased (from 6.9 ± 0.1 µg collagen/mg of tumor to 5.5 ± 0.8 µg collagen/mg tumor; Figure 2d). During the early days, before exponential tumor growth, the segregation of the fibroblasts and the arrival of T cells, the immune infiltrate, is primarily made up of innate immune cells, mainly macrophages [35,44]. Therefore, it was in our interest to understand the interactions and supporting role of fibroblasts in generating conditions that lead to tumor-promoting macrophages and favor tumor growth. 

We first examined spheroids comprised of different combinations of macrophages, fibroblasts and melanoma cells. The cell structure and adhesiveness of the spheroids changed depending on their composition. BMDMs failed to form spheroids when cultured alone, but were able to incorporate them into spheroids when combined with other cell types (Appendix A and Figure 2e). The spheroids comprising BMDMs and YUMM cells (M0+Y) were loose, hollow cell clusters with macrophages interspersed. Adding fibroblasts to the BMDMs (M0+F) generated more solid spheroids but with indistinct boundaries from which cells escaped over time. Notably, spheroids with BMDMs, YUMM cells and fibroblasts (M0+Y+F) exhibited distinct features, forming solid, dense spheroids with well-defined smooth boundaries (Figure 2e), resembling the compact structure of in vivo YUMM tumors [44]. These results support the critical role of melanoma–fibroblast–macrophage interactions in establishing the growing tumor. However, imaging these structures is difficult due to the high cell density, and so in order to study early cell-cell, we proceeded to develop the 3D collagen cultures.

To recapitulate the melanoma TME in vitro, and study the early interactions between macrophages and stromal cells, we generated 3D cultures in 1.5 mg/mL type-I collagen gels (Figure 2f). We combined YUMM cells with the 3T3MEF fibroblast cell line and BMDMs, and analyzed the co-cultures for up to 7 days. For the cell lines used, we compared different seeding densities for optimal growth in this 3D environment, and we selected 6 × 10^5^ 3T3MEF/mL and 2 × 10^5^ YUMM/mL. We set the initial ratio in our co-cultures of BMDM:Fibroblast:YUMM to 12:3:1 (Appendix A). In this co-culture setting, unstimulated BMDMs (M0) displayed a distinct change in morphology and behavior depending on cell distributions in local areas of the gel. Macrophages in close proximity to YUMM cells or fibroblasts are generally longer and display extended protrusions, with their cell body aligned with the direction of the neighboring cells, reaching out to establish direct contact (Figure 2f and Appendix A). Macrophages farther away from other cell populations are mostly round and small, with a morphology similar to mono-cultured macrophages. Furthermore, we observed collagen alignment and thickening in the gels, colocalizing spatially with the fibroblasts, suggesting that they might be pulling collagen bundles and altering the landscape of ECM over time (Figure 2h). Overall, these observations suggest that macrophages are influenced by other cell types and the ECM in the 3D collagen co-cultures.

### 3.3. 3D Collagen Culture Reveals Progressive Macrophage–Stromal Interactions Leading to Cell Activation

Once we determined the optimal density and conditions for each cell type in the 3D collagen-I environment, we characterized the composition of the co-culture at different timepoints. Starting with a composition containing 75% BMDMs (12M:3F:1Y ratio), we observed a rapid increase in the tumor cells and fibroblasts in the first 24 h, most likely due to their higher proliferation rates. By day 3, the macrophages made up less than 20% of the total culture, while the tumor cells and fibroblasts significantly increased in number, comprising more than 80% of the culture system (Figure 3a). This ratio was sustained over time, showing similar proportions after a week (Figure 3a). We next characterized the health and proliferation of macrophages as a function of other cells in the co-cultures. BMDMs cultured alone in the collagen-I matrix, in complete media but without M-CSF or polarizing cues, remained viable for the first 24 h, but did not survive past day 3 (Figure 3b). However, the presence of either fibroblasts or tumor cells within the culture significantly increased their survival, with fibroblasts slightly more efficient than tumor cells at maintaining the viability of the BMDMs (Figure 3b, day 3 timepoint). Consistent with viability, the proliferation of macrophages in the 3D co-cultures gradually increased over time, and by day 7, most of the BMDMs had undergone at least one division cycle (Figure 3c). The presence of fibroblasts induced increased macrophage proliferation as early as day 3, and also supported a larger fraction of highly proliferative macrophages (Figure 3d). Increased BMDM viability and proliferation in 3D co-cultures with tumor cells and fibroblasts was observed with all polarization states (Appendix A).

We also examined how 3T3MEF fibroblasts were affected in the co-cultures with macrophages and tumor cells. This fibroblast cell line expressed several receptors and adhesion molecules commonly found in primary fibroblasts and CAFs, including PDGFRα, TGFβRI, VCAM1 and Podoplanin, but exhibited heterogeneity in the population (Appendix A). 3T3MEF fibroblasts in the 3D co-cultures had a distinct secretion profile from the basal state and TGF-β1 stimulation. Their interaction with the BMDMs and tumor cells in the collagen-I matrix increased production of GM-CSF, G-CSF, IL-6, CCL2, CCL3, CCL4 and CCL12, and reduced production of IFN-β1, LIF, VEGF, CCL17, CXCL5 and CX3CL1 (Figure 3e). We found that 3T3MEFs appeared to be activated by signals from the 3D TME. YUMM-conditioned media alone induced activation of these fibroblasts, observed by morphological changes, upregulation of α-SMA and FAP expression. These fibroblasts were active secretors and contractile in a collagen-I environment for several days, even if alone in the co-cultures (Figure 3f and Appendix A).

To study how the cell components of the 3D culture dynamically interact with the matrix, we quantified the soluble collagen in each condition over time. The presence of different cell types in the co-cultures led to distinct changes in the collagen-I ECM. There was a significant overall loss of soluble collagen between day 3 and day 7 across all conditions, indicating ECM remodeling and collagen degradation (Figure 3g). The polarization state of BMDMs (M0, M1 or M2) did not impact the total soluble collagen present in the 3D co-cultures in the evaluated timepoints (Appendix A). Overall, we conclude that fibroblast activation in the co-cultures was accompanied by the remodeling of the ECM, similar to what we observed in tumors (Figure 2c).

### 3.4. BMDMs Undergo Rapid Major Changes in Morphology and Motility in Response to Melanoma Cells in the 3D In Vitro TME

We next explored the biomechanical activation of BMDMs in 3D co-cultures that could be associated with functional changes in these cells. We quantified parameters related to the size, shape and motility of the BMDMs in 3D co-cultures over time, in response to the presence of tumor cells and/or fibroblasts in the culture. Regarding morphology, we characterized the BMDM compactness (i.e., the degree of protrusive structures that add to the total cell body area) and circularity (i.e., the degree to which cell shape approximates to a circle; see Methods). Most M0 BMDMs were round, with a high compactness index. However, when exposed to tumor cells or fibroblasts, the morphology of BMDMs changed significantly during the first 3 days, and was sustained thereafter (Figure 4a). In co-culture conditions, macrophages elongated and generated protrusive dendrites, although BMDMs from M0+Y+F co-cultures demonstrated a slightly higher frequency of “star-like” morphology and relatively lower compactness (Appendix A). The circularity and elongation of macrophages in co-culture exhibited similar distributions in the first several days (Figure 4b). More heterogeneity in cell shape was observed in co-culture versus mono-culture conditions.

The average speed of macrophage migration in 3D co-cultures with fibroblasts and tumor cells increased over time; in contrast, the average speed of macrophages in mono-cultures remained low (Figure 4c,d). This demonstrates that the presence of melanoma cells and/or fibroblasts, provided cues that not only maintained macrophage survival but also promoted activation and migration. Macrophages in M0+Y and M0+Y+F generally exhibited similar levels of motility, as shown by their average displacement within the 3D gels (Figure 4e). Of note, BMDMs in the M0+Y condition migrated around the culture slightly more than the M0+Y+F condition on day 7, as seen both by average speed and displacement (Figure 4c,e).

We observed high synchronicity between individual macrophage behavior in the co-cultures combining melanoma cells and fibroblasts. Initial activation for the first 24 h induced a non-directional, oscillatory movement in the BMDMs. Then, most macrophages acquired a spindle-like shape on day 3, with a “scouting mode” on, moving faster and longer distances, contacting neighboring cells. Finally, they transitioned into a more stationary state, with star-like morphology on day 7 (Video S1). We did observe dynamic changes in morphology when starting the co-cultures with pre-polarized BMDMs (i.e., M1 or M2; data not shown), and they showed similar migration behaviors to M0, in both mono-culture and co-cultures (Appendix A).

To determine if the activation of macrophages in the 3D environment is dependent on the relative distance to the tumor cells, we tracked the speed of each macrophage relative to its initial distance from a tumor cell (Figure 4f,g and Appendix A). We found that the average speed of macrophages initiated within 100 µm of tumor cells was significantly higher than for macrophages that started further away (Figure 4h and Appendix A). However, proximity to a tumor cell did not guarantee high macrophage motility. Overall, we conclude that the presence of tumor cells and fibroblasts have a strong effect on macrophage motility in the co-cultures.

### 3.5. BMDMs in the In Vitro 3D TME Show an Immunosuppressive Transition into a Melanoma TAM-like Profile

The parameters analyzed above, including direct cell–cell interactions, forces transduced by the 3D matrix and motility, and the membrane-bound or secreted cues, suggested that macrophages would be influenced by these co-culture conditions. Therefore, we evaluated how macrophages evolve in the co-cultures over time. We were particularly interested in assessing the plasticity of their phenotype as the culture progressed, and determining which initial polarization state (M0, M1, or M2) would most resemble TAMs isolated directly from the melanoma TME.

As illustrated previously in Figure 1c, YUMM/ER melanoma TAMs show a complex phenotypic profile, and M2-like immunosuppressive markers are expressed in different proportions (Appendix A). The BMDMs we used to model the melanoma TME can be polarized in vitro into an M2-like state, but not all the markers can be equally induced or solely identify unique populations, like in in vivo compartments (Appendix A). In particular, CD163 showed very low initial expression and did not exhibit changes during the 3D culture but others, like Arg-1 or CD206, were predominantly expressed in M2-polarized conditions and showed remarkable variations across conditions and timepoints (Appendix A). In order to assess the phenotypic evolution of BMDMs in the 3D cultures, we tracked the set of markers that exhibited a robust expression and variation across conditions, and are also present in the melanoma TAMs. This would allow us to compare in vivo and in vitro populations and more accurately establish similarities between them.

We first assessed the phenotype of M0 BMDMs in the 3-cell co-culture (M0+Y+F) conditions using multiplexed flow cytometry. When we visualized these measurements using UMAP, we observed diverse subpopulations of “3D TAMs” that evolved over time (Figure 5a), with a fraction of BMDMs acquiring expression of Arg-1, CD206 and F4/80. After 7 days of co-culture, the phenotypic profile of the M0 BMDMs evolved to look more similar to the average phenotype of the melanoma TAM profile in early (day 7) tumors, as analyzed by PCA (Figure 5b). When analyzing 3D co-cultures with BMDMs cultured with only one other cell type (i.e., with fibroblasts or tumor cells), we found that it was the fibroblasts that favored the M0 BMDM transition, observing a larger fraction positive for Arg-1 by day 7 (19% vs. 13%; Figure 5c). When all three cell types were combined, there was a more dynamic evolution of the phenotype. Initially, a fraction of M0 BMDMs acquired iNOS expression, which decreased after day 1, while rapidly acquiring Arg-1 expression (both iNOS^+^ and iNOS^−^ at day 1), and increasing towards day 3, as compared to the 2-component cultures (Figure 5c).

Interestingly, when BMDMs were pre-polarized with an M1 or M2 phenotype, they exhibited different trajectories in their temporal phenotypic evolution in the co-cultures that were distinct from the mono-cultures, but were also less similar to the TAMs than BMDMs initialized in an M0 state, as analyzed by PCA (Appendix A). A more detailed examination of their evolution revealed that BMDMs retained characteristics of their starting states up to day 3; and between days 3 and 7, BMDMs started expressing complex phenotypes with mixed M1/M2 markers regardless of their initial starting state (Appendix A). Interestingly, M1 BMDMs showed a strong dependence on the polarizing cues to maintain their profile, as evidenced by their rapid loss of iNOS expression in the co-cultures. In contrast, M2 BMDMs exhibited a stronger commitment to their induced phenotype, with increasing expression of Arg-1 in the co-cultures (Appendix A).

We further examined co-cultures initialized with a mix of BMDM polarization states, as might be expected to occur in a more complex tissue at the point of tumor initiation. As shown previously, YUMM TAMs exhibited a mixed M1/M2 profile, though not as clearly polarized as reference BMDMs (Figure 1a–c and Figure 5d). Starting with different M0:M1:M2 ratios in the macrophage fraction of the co-cultures generated by day 7 more diverse “3D TAM” populations (Figure 5d). Mixed BMDMs in the co-cultures maintained characteristics of their initial states through day 3, and seemingly evolving independently from one another. However, by day 7, the BMDMs were expressing mixed M1/M2 markers similar to the M0 BMDMs evolved in the co-cultures (Appendix A).

Finally, to evaluate how closely our “3D TAMs” resembled more advanced tumor stages, we compared BMDMs initialized in either the M0 state or in a mix of M0:M1:M2 states, followed by evolution in the 3D co-cultures, and compared them to d7 and d14 melanoma TAMs. We chose to compare both timepoints since we observed heterogeneity between TAM samples collected at the same timepoint and we had observed a dynamic TAM phenotype in vivo (Figure 1a). We observed that “3D TAMs” initialized in either the M0 or the mixed state evolved towards TAMs isolated from YUMM tumors, as analyzed by PCA (Figure 5e). Interestingly, when examining the heterogeneity of the TAMs using UMAP, we found that “3D TAMs” clustered slightly closer to d14 YUMM TAMs, displaying the expression of a combination of immunosuppressive and pro-inflammatory markers (Figure 5f,g and Appendix A). In addition, it appeared that BMDMs initialized in a mix of states clustered closer to YUMM TAMs than a pure M0 culture (Appendix A). Overall, we conclude that co-cultures with tumor cells and fibroblasts shape BMDM phenotypic profiles regardless of the initial state, but initializing with a mix of M0:M1:M2 BMDMs, and to a lesser extent with M0 BMDMs, leads to a phenotype most similar to TAMs from YUMM/ER tumors.

### 3.6. BMDMs in the In Vitro 3D TME Acquire a Sustained Mixed M1/M2-like Functional Profile Resembling Melanoma TAMs

Lastly, we evaluated the secretion profile induced in BMDMs evolved in the 3D TME co-cultures as a measure of the potential effector functions of “3D TAMs” and the other cell components. We analyzed co-cultures containing all the cell types but varied the initial composition of polarized BMDMs (i.e., M0, M1, M2, or mixed). After 7 days in the 3D co-cultures, we extracted and sorted the three cell types, cultured them to collect their conditioned media and then analyzed the secreted C/Cs with a multiplexed immunoassay (Figure 6a). We observed that “3D TAMs” secreted higher levels and a broader range of C/Cs on the panel, as compared to tumor cells and “3D CAFs”. Some of these secreted proteins were shared with the “3D CAFs”, such as CXCL1, VEGF, G-CSF and GM-CSF. The tumor cells secreted a few C/Cs on the panel, including exclusive secretion of IL-11, and shared secretion of CX3CL1, LIF and TIMP-1 with the fibroblasts (Figure 6b).

Similar to the results from our phenotypic analysis, the 3D TME shaped the BMDM secretory profile to resemble the one observed in melanoma TAMs. PCA analysis of the secretion profile showed a transition path similar to our phenotypic analysis, in which secretion was initially aligned with polarized BMDMs, but then transitioned to a more complex state resembling TAMs from YUMM tumors (Figure 6c). Coculturing BMDMs with tumor cells and fibroblasts in 3D conditions induced a secretion profile that exhibited a similar pattern to TAMs, with some of these complex M1/M2 features, after only 7 days (Figure 6d,e). When looking in detail at the multiplexed secretion, the 3D TAMs secreted TNF-α, IFN-β1 and CXCL10, as well as IL-10, CCL17 and CCL22. This 3D TAM profile correlated with the standard M2-like profile, to a similar degree as TAMs (Figure 6f). However, the co-cultures established with BMDMs that were M2 pre-polarized only, rapidly changed the production of these cytokines over time (Appendix A). However, not all the networks from the melanoma TME were reproduced in vitro. Supernatant from whole YUMM tumors contained a significant number of shared cytokines/chemokines, but also contained IL-1β, IL-2, IL-4, IL-9, IL-17, CCL20, CCL21, among others, which were not observed in the co-cultures and so were most likely produced by other components not included in vitro, such as other immune cells or endothelial and lymphatic cells (Appendix A).

## 4. Discussion

It is increasingly clear that in vitro macrophage polarization protocols do not re-capitulate the complex macrophage phenotypes observed in vivo [45,46]. Single-cell approaches have provided a more complete look into macrophage heterogeneity in tumor infiltrates [47,48], with surrounding cells and cues that influence their behavior and shape their functional response [12,49,50]. Developing experimental models to study TAMs in vitro is not as simple as inducing a cell program through a signaling pathway, especially when the definition of polarization state, such as the M2-like profile, has expanded over the years to fit the pathophysiology of the macrophages [26]. TAMs in the YUMM model were no exception, with strong heterogeneity and an evolving profile over time. Single-cell analysis showed that individual melanoma TAMs expressed different levels of multiple markers associated with M2-like states, like Arg1, CD206 and CD163, usually co-expressed, but also defining individual subsets (Appendix A). Macrophages embedded in the TME also had a mixed M1/M2 phenotype, with patterns of co-expression of markers such as CD206 and MHCII, and iNOS and Arg1. Even though some of these individual markers are associated with a pro-inflammatory M1-like profile, they have also been shown to favor immune suppression [51,52]. On the other hand, the secretion profile revealed an intriguing pattern with cells displaying polyfunctionality (i.e., secreting proteins associated with both M1 and M2 states), complicating their classification. Additionally, a majority of TAMs presented as resting cells or low secretors, suggesting a potential to be re-educated towards favoring an effective anti-tumor response [21,53,54]

Interestingly, we found that TAM heterogeneity is lost when isolated and cultured ex vivo in 2D conditions, emphasizing the need for TME signals to maintain their identity. In recent years, 3D cultures in hydrogels, including collagen [55,56], have provided valuable tools to accelerate tumor studies and TME modeling [33,34]. In the present study, we developed a 3D collagen co-culture system to mimic the melanoma TME and investigate how interactions between melanoma cells, fibroblasts, and macrophages shape the early stages of macrophage immune activity leading to such complex phenotypes. At these early timepoints, immune infiltration in the tumors is primarily from myeloid lineages, with tissue-resident and newly-recruited components. While our 3D system does not recapitulate all influential immune components, such as dendritic cells [57], there is strong evidence that macrophages play a key role in generating a pro-tumoral niche [3,11]. Here, we found that with a relatively simple 3-cell type reconstruction of the TME, we were able to capture a macrophage immunosuppressive transition and see them acquire a phenotypic and functional profile similar to that of TAMs isolated directly from growing melanoma tumors.

The 3D 3-cell type reconstruction of the TME provided survival, proliferation and activation cues to the macrophages, building evident cell circuits with the fibroblasts, as described before [58]. However, by starting with extreme polarization state profiles (M0, M1, and M2), we identified different trajectories in which the macrophages evolved their phenotypic and functional profiles. In an effort to recreate a starting point that would ultimately lead to the melanoma TAM-like state, we generated conditions in which macrophages expressed markers and secreted factors associated with both M1 and M2 polarization (Figure 5). Interestingly, just like our 3D TAMs, the M2 macrophage profile partially correlated with the overall TAM phenotypic profiles, reinforcing the traditional idea of their categorization as immunosuppressive. However, a detailed analysis of the markers indicates that there still was a mismatch on M1-associated markers. With this 3D TME, we were able to generate a stable system that generated and combined most of these M2-like immunosuppressive features with M1-like parameters, and was sustained over time.

Evidence from our work showed that adding fibroblasts to macrophage + tumor cells limits macrophage migration after day 3, most likely related to ECM remodeling. It remains to be studied whether this has an impact on the dynamics in vivo. When YUMM tumors start to grow exponentially after days 5 and 7, they may be leaving fibroblasts physically excluded in the periphery, with a higher ratio of tumor cell to fibroblast, in line with our measured reduction of total collagen per tumor mass. This newly-formed surrounding fibroblast belt could provide a compact tumor structure, as we have seen in the in vitro YUMM spheroids, and potentially generate a physical barrier with additional immunosuppressive signals to the arriving T cells [59].

From the time-lapse imaging in our 3D co-culture system, we observed several significant trends in macrophage behavior over time. First, we found that the average speed of macrophage migration increased inversely with the distance to the closest YUMM melanoma cell (Figure 4). A previous study suggested that dynamic fibroblast contractions can recruit macrophages when co-cultured in fibrillar collagen [20]. YUMM cells likely activate macrophages in a similar way by dynamically contracting surrounding collagen. Second, we observed that macrophages in the co-cultures increased their average speed and became more stellate-like over time, which might be in part due to dynamic changes in the collagen environment. Both macrophages and fibroblasts are highly active in remodeling collagen [60,61,62]. In addition, macrophages are plastic and respond to material surfaces or collagen architecture by adopting different shapes [13,63,64]. Macrophage shape has been linked to phenotype, as elongation of macrophages without external cytokine stimulation was shown to induce markers of M2 polarization [13]. In our system, we also observed that elongation of macrophages was associated with an M2-like evolution. Direct contact with neighboring tumor cells and/or fibroblasts through integrins can also shape macrophage phenotype [65]. Importantly, by day 5, macrophages in the 3D co-cultures resembled the morphology of TAMs observed via intravital imaging [29,66].

In addition to biomechanical cues, it is likely that BMDMs in the co-cultures are responding to biochemical cues in the microenvironment. Fibroblasts and macrophages create a stable circuit in tissues through the exchange of growth factors [58]. In our co-cultures, we observe fibroblasts secreting several growth factors and cytokines with influential roles in the TME. In the co-cultures, G-CSF and GM-CSF could account for the viability and proliferation of macrophages, in addition to their own production of M-CSF [67], as well as fractalkine/CX3CL1, which has been implicated in anti-cancer responses [68].

The complex 3D environment was strongly influenced by cell composition, suggesting that combining BMDMs with fibroblasts and tumor cells elicited cellular cross-talk that impacted the overall presence of immune modulators in the 3D TME. These changes in the microenvironment could be seen after only one day of co-culture. The M0+F+Y conditioned media showed increased levels of M-CSF, G-CSF, IFN-β1, CCL2, CCL5, CX3CL1, and slightly more IL-6, CCL3 and CCL4, when compared to M0+Y and M0+F conditions (Appendix A). Chemokines have been reported to have a role in influencing macrophage polarization [69]. Some of these chemokines, such as CX3CL1 or CCL5 and CCL2 have been reported to enhance the alternatively activated profile, specifically with the upregulation of Arg-1 [70,71,72], which would explain the induction of this marker only in BMDMs cultured in the complete 3D TME. Macrophages shaped in the co-cultures acquired the capacity to secrete pro-inflammatory cytokines associated with an M1 polarization state (e.g., TNF-α, IL-6, and CCL3), as well as factors more commonly associated with an M2 phenotype (e.g., CCL17, CCL22, and IL-10). This reflects what we previously measured from TAMs isolated from in vivo tumors (Appendix A) [21], and makes this system a valuable tool to study cell–cell interactions in the early TME with the potential to evaluate changes induced by targeting specific cross-talk pathways to break immune suppression.

As discussed above, this is a closed system, which is limited to the original components used to set up the culture and is further limited in the total time the culture can be monitored without collapsing. In this 3D TME, we are able to evaluate local responses of macrophages and fibroblasts to the addition of perturbations to the system. The possible next steps would include sequentially incorporating other cell types in the 3D TME, as shown recently [33], and also capturing some features of this dynamic transition in live tumors and interaction with other tissue structures and cell types, using lineage and functional reporters [73,74]. Live-tumor experiments would also allow us to understand better the dynamics of the exclusion of fibroblasts. During these first 7 days, melanoma cells interact with fibroblasts and macrophages to build and establish their immunosuppressive, tumor-promoting TME.

## 5. Conclusions

We developed an in vitro 3D collagen model of cell–cell interactions that shape immune activity in the early melanoma TME. In this in vitro model, stromal cells induced increased motility and response from macrophages, and macrophage acquired a pheno- type and functional profile that resembled TAMs from melanoma tumors. Overall, in this 3D system, the cell components rapidly generated cell circuits that built an environment capable of inducing an immunosuppressive functional signature in the macrophages, as observed in vivo. This transition was fine-tuned by varying initial macrophage states, capturing several complex and diverse features of melanoma TAMs, without external perturbations. During these first 7 days, melanoma cells interact with fibroblasts and macrophages to build and establish their immunosuppressive, tumor-promoting TME. Our system will enable future studies of changes in macrophage–stromal cross-talk in the melanoma TME.

## Figures and Tables

**Figure 1 cancers-13-04579-f001:**
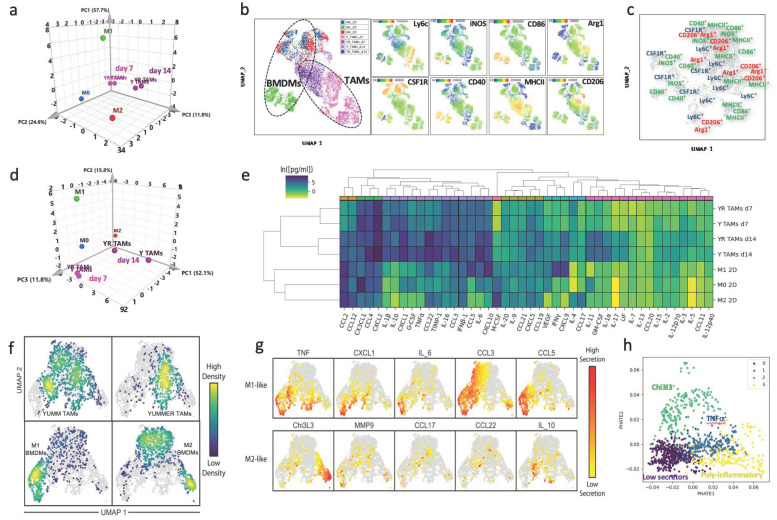
Characterization of the phenotype and functional profile of melanoma TAMs. (**a**) Phenotypical profile of melanoma TAMs. Principal component analysis (PCA) of the phenotype of in vitro stimulated BMDMs compared to TAMs from YUMM and YUMMER early (day 7) and late (day 14) tumors. This phenotypical characterization was assessed by flow cytometry using a panel of 10 markers and then combining the percentage and MFI of those parameters for the PCA analysis. For BMDMs *n* = 5, TAMs *n* = 4/5. (**b**) UMAP plots containing representative FACS single-cell data from a. Left, sample annotation. Right, detailed panels with heatmaps showing the expression of different markers used for analysis. (**c**) UMAP visualization with superimposed annotation of markers expressed in different regions of the phenotypic space in YUMM d14 tumors. (**d**) Functional profile of melanoma TAMs. PCA of the bulk secretion of in vitro stimulated BMDMs compared to TAMs from YUMM and YUMMER early and late tumors. (**e**) Hierarchical clustering analysis of the secretion profile of Y and YR TAMs at early and late timepoints compared to BMDM reference polarized states. (**f**) UMAP visualization of single-cell secretion profiles combining Y and YR TAMs with in vitro polarized BMDMs into M1 or M2. (**g**) Detail of the protein secretion associated with and M1- or M2-like function and distribution between conditions. (**h**) Annotation of the TAM functional clusters from YUMM and YUMMER tumors, in the PHATE plots, showing the generation of a pro-inflammatory axis on the first component and a Chi3l3 immunosuppressive axis on the second component.

**Figure 2 cancers-13-04579-f002:**
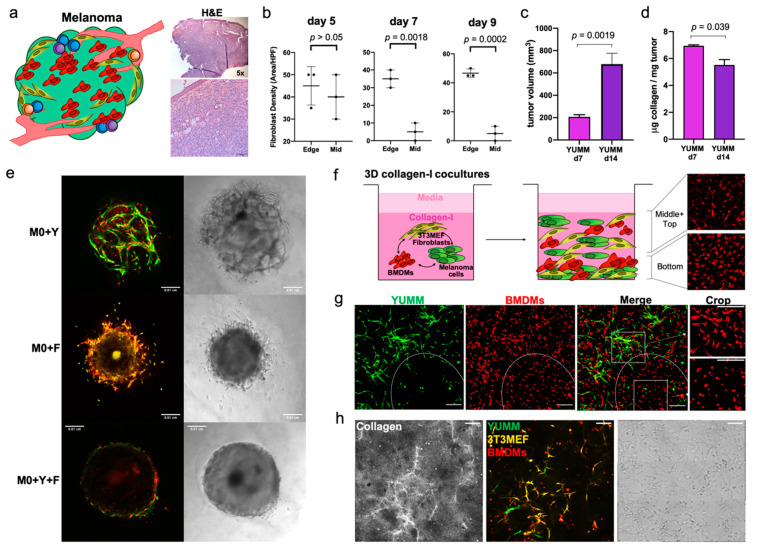
Mimicking the early melanoma TME with a 3D co-culture system. (**a**) Characterization of YUMM tumors. Schematic representation of the melanoma microenvironment (left) and H&E staining of tumors showing the intradermal tumor structure, infiltrating the whole dermal compartment and the subcutaneous space (right). (**b**) Quantification of EGFR^+^ fibroblast density in two regions of the tumor, the middle (Mid) and the Edge. In this melanoma model, fibroblasts are evenly distributed across regions of early tumors but, after day 7, they are rapidly segregated to the edge. (**c**) Change in tumor volume between d7 and d14 in YUMM (*n* = 6). (**d**) Quantification of soluble collagen in YUMM tumors (d7 *n* = 2, d14 *n* = 4). (**e**) Representative images of the morphology of spheroids of different compositions, embedded in collagen matrix. Y = green, M0 = red, F = yellow. (**f**) Schematic of the proposed translation of the melanoma TME into a 3D collagen cultures for in vitro analysis, combining BMDMs with YUMM cells and fibroblasts. After a few days in these cultures, BMDMs show different activation states, depending on their and the presence of other cells. (**g**) Representative image of a YUMM+BMDM 3D culture seeded as a single cell suspension of the cells. The white ellipse delineates an area where macrophages are located further away from YUMM cells. Inserts on the right are cropped areas delineated by the dotted squares. Zoomed-in images of how macrophages differ in morphology as soon as 24 h after starting the co-culture, Scale bar in 200 µm. (**h**) Representative images of collagen remodeling at 48 h post gel embedding, containing unstimulated BMDMs (M0), 3T3MEF fibroblasts and YUMM cells. Initial cell seeding number M0: 24 × 10^3^/well, YUMM: 2 × 10^3^/well. Fibroblast: 6 × 10^3^/well, in 10 µL gel. Scale bars are 100 µm.

**Figure 3 cancers-13-04579-f003:**
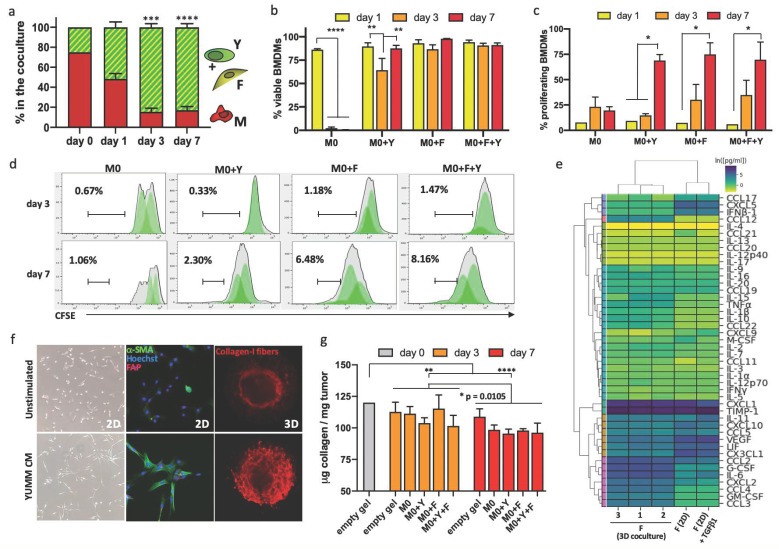
Rapid functional cell activation after early interactions in the 3D melanoma TME. (**a**) Evolution of the composition of cell types in 3D cultures over time, starting from a 12:3:1 M-F-Y ratio (d1 *n* = 4, d3 *n* = 5, d7 *n* = 8). (**b**) M0 BMDM survival in 3D cultures over time. Viability of macrophages was assessed after being cultured alone or in combination with the other cell components, at d1, d3 and d7 (*n* = 3). (**c**) M0 BMDM proliferation in 3D cultures overtime. We analyzed cell divisions of macrophages after CFSE staining, and considered in the proliferating fraction all cells that had undergone at least one division cycle (*n* = 2). (**d**) Representative histograms of macrophage proliferation from all the co-culture conditions, gates are highlighting the high proliferative subpopulations (over two divisions at each timepoint). (**e**) Hierarchical clustering of 43-plex secretion profile analysis of 3T3MEF fibroblasts. We compared secretion in resting cells with TGF-*β*1 stimulation and fibroblast from 7d 3D co-cultures. (**f**) Activation of the 3T3MEF cell line by YUMM melanoma cells. Fibroblasts were stimulated with YUMM conditioned media (CM), and we observed morphological changes under the light microscope and activation through the expression of *α*-SMA. When cultured in 3D cultures, we assessed fibroblast functional activation in a stained collagen pulling assay. (**g**) Quantification of total soluble collagen-I in the M0 3D cultures, at d3 and d7, compared to the initial collagen-I added in each. A control well with no cells was kept as a control of the quantifiable soluble collagen-I (*n* = 4). * *p* < 0.05, ** *p* < 0.01, *** *p* < 0.001, **** *p* < 0.0001.

**Figure 4 cancers-13-04579-f004:**
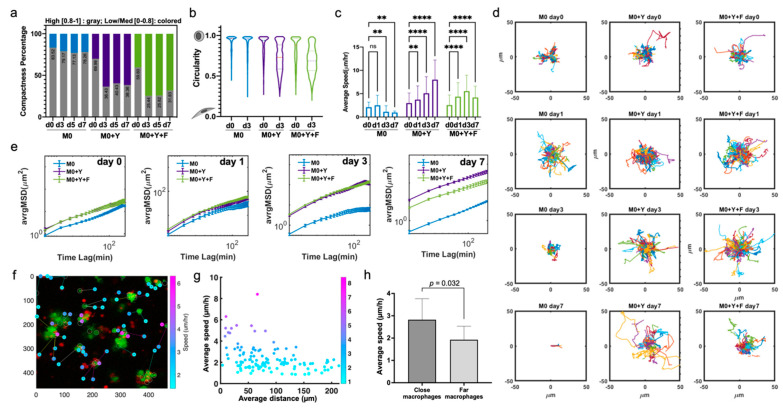
Significant morphology and motility changes in BMDM activation in 3D co-cultures over time. (**a**) Generation of cell protrusions in M0 BMDMs across multiple days exposed to different 3D environments. Compactness was calculated as described in the methods. Cells were first classified into two categories, low/medium compactness [0–0.8] and high compactness [0.8–1]. For each condition and time, proportion of cells in each category were calculated and plotted as percentages of total. Gray bar, high compactness [0.8–1], colored bar, low/med compactness [0–0.8]. (**b**) Violin plots showing the early changes in BMDM morphology in different 3D environments. Circularity index was calculated as described in MM. (**c**) Average migration speed of macrophage, data shown as Mean +/− SD. (**d**) Overlaid individual migration trajectories of M0 BMDMs in different 3D environments, at different timepoints. (**e**) Average mean square displacement (avrgMSD) of M0 BMDMs across multiple days exposed to different 3D environments. data shown as Mean +/− SEM. (**f**) Representative image of calculation of distance between macrophages and the closest tumor cell at early timepoint (day 1). Overlaid data on individual average speed of the BMDM. (**g**) Scatter plot of M0 BMDMs average speed vs. distance to the closest YUMM cell after 24 h of 3D co-culture. (**h**) The bar plots compare the average speed of the top 15 closest and top 15 farthest macrophages from the nearest YUMM cell at day 0. ns: Not significant, ** *p* < 0.01, **** *p* < 0.0001.

**Figure 5 cancers-13-04579-f005:**
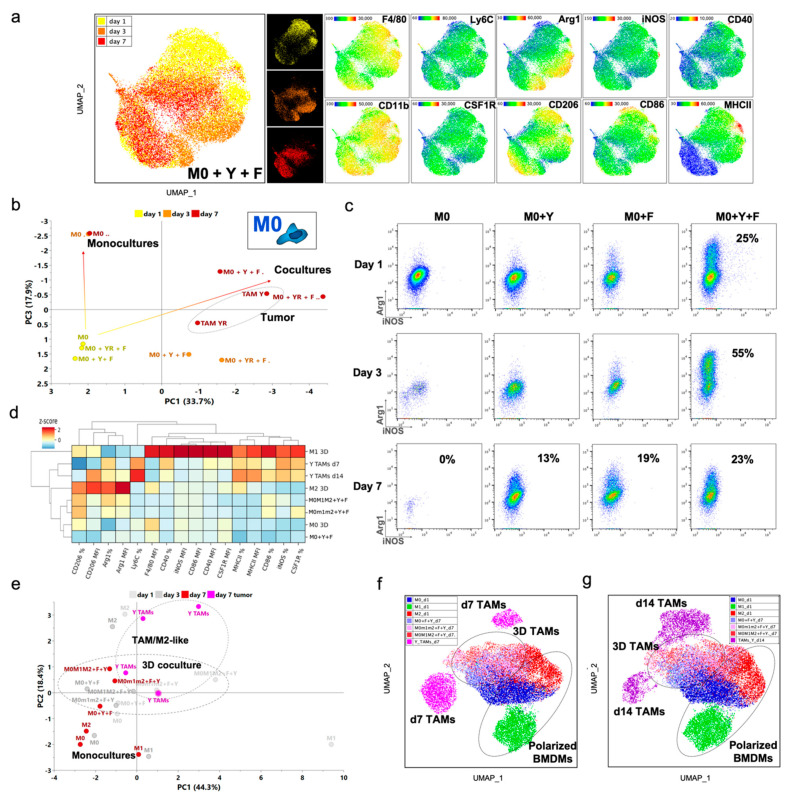
Evolution and modeling of BMDM phenotype into a TAM-like state. (**a**) Analysis of unstimulated BMDM (M0) phenotype over time in 3D co-cultures. UMAP clustering showing macrophages from the M0+F+Y co-culture at d1, d3 and d7 combined. Heatmaps provide detailed distribution of marker expression, highlighting different subpopulations and evolution over time. (**b**) PCA analysis of M0 BMDM phenotype trajectory over time in 3D cultures, compared to day 7 melanoma TAMs. (**c**) Representative dot plots of M1/M2 markers iNOS/Arg1 in M0 BMDMs as mono-cultures and different combinations of macrophage 3D co-cultures, and their evolution over time. (**d**) Hierarchical clustering of marker z-score of macrophages from polarized BMDM profiles, 3D co-cultures at d7 and early (d7) and late (d14) melanoma tumors. (**e**) PCA analysis of macrophage phenotype over time in 3D co-cultures. Highlighted in color, M0 3D mono-cultures and co-cultures and TAMs, collected on day 7. Dashed circles delineate the phenotypic space by sample type, showing the partial overlap between the YUMM TAMs and in vitro “3D TAMs”. (**f**) UMAP clustering of macrophages from 3D mono- and co-cultures and early (d7) melanoma tumors, comparing single-cell phenotype. (**g**) UMAP clustering of macrophages from 3D mono- and co-cultures and late (d14) melanoma tumors, comparing single-cell phenotype.

**Figure 6 cancers-13-04579-f006:**
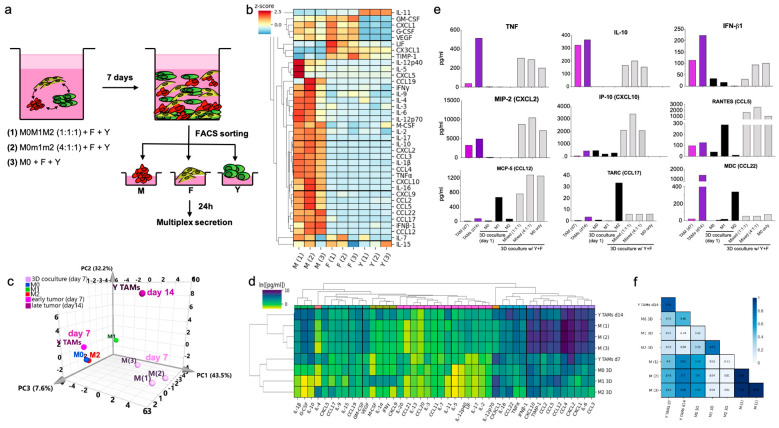
BMDM transition into an immunosuppressive TAM-like functional profile after 7 days of 3D co-culture. (**a**) Schematic representation of the isolation of components after 3D co-culture for secretion profiling combining BMDMs with fibroblasts and YUMM cells. (**b**) Heatmaps with z-scores for all the proteins studied in all three fractions. (**c**) Functional profile of ‘3D TAMs’ compared to melanoma TAMs. PCA of the bulk secretion of macrophages from 3D co-cultures, in vitro stimulated BMDMs and TAMs from YUMM early and late tumors. (**d**) Hierarchical clustering of 43-plex secretion profile analysis of ‘3D TAMs’ and Y TAMs at early and late timepoints compared to 3D BMDM reference polarized states. (**e**) Bar graphs with the absolute concentration of secreted cytokines/chemokines in sorted macrophages from 3D co-cultures at day 7, and compared with polarized reference BMDMs and early and late Y TAMs. (**f**) Heatmap of Pearson correlation of the secretion profiles of BMDM in 3D cultures and d7 and d14 melanoma TAMs. In M0M1M2, polarized BMDMs were mixed in equal parts in the same culture. In M0m1m2, polarized BMDMs were mixed in ratio 4:1:1 in the same culture.

## Data Availability

The data presented in this study are available in Appendix A.

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
