# Peer review of "3D Model of the Early Melanoma Microenvironment Captures Macrophage Transition into a Tumor-Promoting Phenotype"

_cancers, 2021, doi:10.3390/cancers13184579_

Round 1
Reviewer 1 Report
The original research article is potentially interesting and worthy of eventual publication. However, I found some problems that preclude publishing in the present form.
- In the Fig. 5, arg-1 expression in macrophages was increased under 3D culture condition. The authors should describe the reason why Arg-1 is up-regulated in the discussion section.
- (page 8, line 327) The authors described that “We identified functional clusters within these TAMs, which aligned into two activation axes, one having an M2-like profile and another one with a mixed pro-inflammatory one (Figure 1h, Figure S1f).” It should be stated what is considered from these results. Why is this phenomenon observed?
- CD163 is well known to be a major M2 phenotypic marker as well as CD206; results on CD163 expression should also be presented.
- Arg-1 is known to be an unsuitable M2 phenotype marker in human macrophages. The fact that phenotype markers are different between human macrophages and mouse macrophages should also be mentioned in the manuscript.
Author Response
[Reviewer comments] The original research article is potentially interesting and worthy of eventual publication. However, I found some problems that preclude publishing in the present form.
[Reviewer comment #1] In the Fig. 5, arg-1 expression in macrophages was increased under 3D culture condition. The authors should describe the reason why Arg-1 is up-regulated in the discussion section.
[Response #1] We thank the reviewer for noting the somewhat surprising result displayed in Fig. 5c, showing that Arg-1 protein expression is increases significantly in the 3D cultures of M0+F+Y, as compared to the M0+Y or M0+F conditions, after only 1 days in culture. To build hypotheses about the underlying causes of this increase, we considered the changes in cytokine/chemokine secretion in the 3D cultures of M0+F+Y by day 1 that are distinct from the M0+Y or M0+F, and that could lead to the upregulation of Arg-1 based on evidence gathered from existing literature reports. We note that the M0+F+Y show more M-CSF, G-CSF, IFNb1, CCL2, CCL5, and CX3CL1 when compared to either M0+Y or M0+F. Some of these chemokines, such as CX3CL1 or CCL5 and CCL2 have been reported to enhance the alternatively activated profile, specifically with upregulation of Arg1 (e.g., PMID: 30245686, PMID: 29166611, PMID: 26254342). Thus, we hypothesize that combining these three cellular components in the cultures is generating cell-cell communication that rapidly shapes the 3D TME and promotes the upregulation of specific immunosuppressive markers in the M0 BMDMs. We have added a paragraph summarizing this evidence to the Discussion (line 682-699).
[Reviewer comment #2] (page 8, line 327) The authors described that “We identified functional clusters within these TAMs, which aligned into two activation axes, one having an M2-like profile and another one with a mixed pro-inflammatory one (Figure 1h, Figure S1f).” It should be stated what is considered from these results. Why is this phenomenon observed?
[Response #2] We thank the reviewer for pointing out that our conclusions from these observations were not clear. Overall, based on both our bulk and single-cell secretion data, we conclude that melanoma TAMs from growing tumors exhibit mixed M1-M2 profiles, i.e., that have characteristics of both the M1 and M2 polarization states generated from simple in vitro cultures. We have clarified these findings in the Results section by discussing the mixed single-cell profiles in more detail (see lines 339-352) and in the Discussion as well (lines 606-617).
[Reviewer comment #3] CD163 is well known to be a major M2 phenotypic marker as well as CD206; results on CD163 expression should also be presented.
[Response #3] We thank the reviewer for raising this point and we have explored it in more detail in this revision. Although CD163 has been used in therapeutic strategies for targeting immunosuppressive TAMs (PMID: 32752088, PMID: 31375534), we observe low CD163 expression in BMDMs, and thus it is not as robust of an M2-marker in our system when compared to Arg-1. We also did not see much expression of CD163 in the tumors as compared to Arg-1 or CD206 (see new panels in Fig. S5a-c). We have added a paragraph to the Results section discussing these findings (lines 501-513).
[Reviewer comment #4] Arg-1 is known to be an unsuitable M2 phenotype marker in human macrophages. The fact that phenotype markers are different between human macrophages and mouse macrophages should also be mentioned in the manuscript.
[Response #4] We agree with the reviewer’s broad point that it is important to acknowledge the challenge of defining M2 macrophage markers due to their variation between species and tumor types. We have added text in the introduction to highlight this important point (lines 71-74). We note that because our work is focused on reproducing the conditions we see in murine melanoma tumors, we determined that Arg-1 was a reliable ‘reporter’ of a broader M2-like state, which we defined through a combination of other phenotypic markers and secretion profiles. Our findings are consistent with findings in other murine tumor models, in which Arg-1 has been used as a marker to identify subpopulations of immunosuppressive cells (e.g., TAMs and MDSCs) (PMID: 30613266, PMID: 33391974, PMID: 23454751). We also note that there is conflicting evidence regarding the expression of Arg-1 in different human tumor types, and recent work suggests that Arg-1 could be targeted for anti-tumor therapy in some tumor types (PMID: 34031449, PMID: 29254508, PMID: 30728077). We think that translating our 3D system to human cell components will be an important next step in evaluating some of these differences.
Reviewer 2 Report
The data presented in the article authored by Pizzurro et al regarding the dynamic of cytokine secretion by tumor associated macrophages following interactions with both tumor and stromal cells is new and provides valuable insights into the role of TME in shaping the anti-tumor immune response in melanoma. The study was well designed and state-of-the-art methods were used to obtain and analyze the data. The results of the present study will be of great interest for the Cancer's journal readers.
The manuscript needs a brief read through: Line 122: please state what type of IFN was used for BMDM polarization.
Author Response
[Reviewer comments] The data presented in the article authored by Pizzurro et al regarding the dynamic of cytokine secretion by tumor associated macrophages following interactions with both tumor and stromal cells is new and provides valuable insights into the role of TME in shaping the anti-tumor immune response in melanoma. The study was well designed and state-of-the-art methods were used to obtain and analyze the data. The results of the present study will be of great interest for the Cancer's journal readers.
The manuscript needs a brief read through: Line 122: please state what type of IFN was used for BMDM polarization.
[Response] We appreciate the reviewer’s positive comments. The manuscript has been thoroughly edited and missing details like the ones raised by the reviewer have been added and/or corrected.
Round 2
Reviewer 1 Report
The Authors have addressed my comments and the manuscript has been improved.